# Gauge theory for topological waves in continuum fluids with odd viscosity

Keisuke Fujii[1,2,*], Yuto Ashida[2,3],

**1** Department of Physics, University of Tokyo, 7-3-1 Hongo, Bunkyo-ku, Tokyo 113-0033, Japan
**2** Department of Physics, Institute of Science Tokyo, Ookayama, Meguro-ku, Tokyo 152-8551, Japan
**3** Institute for Physics of Intelligence, University of Tokyo, 7-3-1 Hongo, Bunkyo-ku, Tokyo 113-0033, Japan
*fujii[at]phys.sci.isct.ac.jp

November 6, 2024

## Abstract

We consider two-dimensional continuum fluids with odd viscosity under a chiral body force. The chiral body force makes the low-energy excitation spectrum of the fluids gapped, and the odd viscosity allows us to introduce the first Chern number of each energy band in the fluids. Employing a mapping between hydrodynamic variables and U(1) gauge-field strengths, we derive a U(1) gauge theory for topologically nontrivial waves. The resulting U(1) gauge theory is given by the Maxwell-Chern-Simons theory with an additional term associated with odd viscosity. We then solve the equations of motion for the gauge fields concretely in the presence of the boundary and find edge-mode solutions. We finally discuss the fate of bulk-boundary correspondence (BBC) in the context of continuum systems.

# 1 Introduction

The topology of the band structures has been established as one of the most fundamental concepts in modern physics [1]. Topological properties are defined as invariance under continuous deformations and are characterized by topological numbers. Owing to the discrete nature of topological numbers, they allow us to classify the phases of matter into distinct equivalent classes. This notion of the topological phases has been originally developed in solid-state physics [2–7] and now extended to a variety of systems, including classical periodic systems [8–10] and non-Hermitian systems [11, 12].

The bulk-boundary correspondence (BBC) is arguably one of the most fundamental properties in the topological phenomena. Namely, when a boundary is introduced to states with a nonzero topological number, there must appear localized modes at the boundary. These localized modes propagate in one direction along the boundary, thus referred to as chiral edge modes. More precisely, the BBC ensures that the net number of the chiral edge modes, which is the difference between the number of modes running in one direction and those running in the opposite direction at the boundary, is equal to a topological number evaluated in an infinitely extended bulk. While the BBC has been shown both theoretically and experimentally for spatially periodic systems such as electronic materials [13–18], it is yet to be established in continuum systems except several cases [19, 20].

In recent years, it has been pointed out that waves arising at boundaries of two-dimensional fluids under external chiral forces, such as the Coriolis or Lorentz forces, can be understood as topological waves akin to chiral edge modes. These topological waves have been discussed in active fluids [21] and in the oceans on Earth [22], attracting interest in the application of the band topology to continuum systems. As a crucial difference from solid crystals, continuum fluids lack the spatial periodicity. Accordingly, unlike a Brillouin zone, the wavenumber space of continuum fluids is not compactified, and thus band structures are not allowed to have well-defined topological numbers.

Nevertheless, a previous study showed that odd viscosity, also known as Hall viscosity, can act as a short-distance regulator in fluids and recover the quantization of a topological number [23]. Here, odd viscosity is a viscosity coefficient corresponding to the Hall response to velocity gradients [24] and arises in fluids with broken time-reversal and inversion symmetries [25–27]. It has been studied in superfluid $^3$He [28–31], magnetic plasmas [32], quantum Hall fluids [33], and more recently in chiral active fluids [34–37], where the self-spinning objects can lead to topological phenomena that are difficult to realize in conservative systems [38, 39]. The BBC for the well-defined bulk topological number associated with odd viscosity in continuum fluids remains an active area of discussion [40–44].

This study aims to describe a continuous fluid having a well-defined topological number within a field-theoretical framework, in hopes that this approach will clarify its topological properties and the BBC. To achieve this, we derive a gauge theory for the linearized hydrodynamic equations that includes odd viscosity, by using the matching between hydrodynamic variables and gauge-field strengths. As we will show, the gauge theory for topologically nontrivial waves is found to be a Maxwell-Chern-Simons theory with an additional term associated with the odd viscosity. While chiral edge modes naturally emerge in the Chern-Simons theory, the role of odd viscosity for the edge modes has yet to be fully explored. In this study, we analyze the edge modes in the Maxwell-Chern-Simons theory

with the odd viscosity term and discuss the BBC in the context of continuous systems.

## 2   Two-dimensional fluids with odd viscosity

### 2.1   Hydrodynamic equations

Let us start with a brief review on topological waves in continuum fluids. We consider linearized Euler equations with odd viscosity in two dimensions:

$$\partial_t \rho(t, \boldsymbol{x}) + \rho_0 \partial_i v_i(t, \boldsymbol{x}) = 0, \tag{1}$$

$$\partial_t v_i(t, \boldsymbol{x}) + \frac{c^2}{\rho_0} \partial_i \rho(t, \boldsymbol{x}) - \nu_o \epsilon_{ij} \nabla^2 v_j(t, \boldsymbol{x}) = \Omega \epsilon_{ij} v_j(t, \boldsymbol{x}), \tag{2}$$

where implicit summations with respect to a pair of repeated indices $i, j, \ldots \in \{x, y\}$ are taken with the Laplacian $\nabla^2 = \partial_i \partial_i$. Here, $\rho(t, \boldsymbol{x})$ and $v_i(t, \boldsymbol{x})$ are hydrodynamic variables, representing the density fluctuation around its mean value $\rho_0$ and the fluid velocity, respectively, and $c$ is the sound velocity. We have introduced the odd viscosity $\nu_o$ per the mean density to discuss topological properties of the fluid in a well-defined manner. The right-hand side of Eq. (2), i.e., $\Omega \epsilon_{ij} v_j(t, \boldsymbol{x})$, represents the chiral body force.

These hydrodynamic equations describe a broad range of systems, such as Hall fluids, rotating fluids, and chiral active fluids. In Hall fluids, which are composed of electrons with a background magnetic field, the chiral body force results from the Lorentz force with $\Omega$ being its cyclotron frequency [45]. In rotating fluids, which are subject to a global rotation, the chiral body force results from the Coriolis force, and accordingly, $\Omega$ corresponds to two times the frequency of the background rotation. Chiral active fluids have an active torque and are described as rotating fluids due to spontaneous rotation of the fluids [35]. Note that shallow water is described by the same equations with $\rho(t, \boldsymbol{x})$ representing the fluctuations of the height [46].

To express Eqs. (1) and (2) in the matrix form, we introduce a vector

$$\boldsymbol{\psi}(t, \boldsymbol{x}) = [\rho(t, \boldsymbol{x})/\rho_0, v_x(t, \boldsymbol{x})/c, v_y(t, \boldsymbol{x})/c]^T, \tag{3}$$

and its Fourier transform $\boldsymbol{\varphi}(t, \boldsymbol{k}) = \int d^2 x \, e^{i \boldsymbol{k} \cdot \boldsymbol{x}} \boldsymbol{\psi}(t, \boldsymbol{x})$. Then, Eqs. (1) and (2) in the Fourier space read $-i \partial_t \boldsymbol{\varphi}(t, \boldsymbol{k}) = H(\boldsymbol{k}) \boldsymbol{\varphi}(t, \boldsymbol{k})$ with $H(\boldsymbol{k}) = \boldsymbol{h}(\boldsymbol{k}) \cdot \boldsymbol{L}$ and $\boldsymbol{h}(\boldsymbol{k}) = [c k_x, c k_y, \Omega - \nu_o k^2]^T$. Here, $\boldsymbol{L} = [L_x, L_y, L_z]^T$ are $3 \times 3$ matrices given as

$$L_x = \begin{bmatrix} 0 & 1 & 0 \\ 1 & 0 & 0 \\ 0 & 0 & 0 \end{bmatrix}, \; L_y = \begin{bmatrix} 0 & 0 & 1 \\ 0 & 0 & 0 \\ 1 & 0 & 0 \end{bmatrix}, \; L_z = \begin{bmatrix} 0 & 0 & 0 \\ 0 & 0 & -i \\ 0 & i & 0 \end{bmatrix}, \tag{4}$$

and satisfy the SO(3) algebra, i.e., $[L_x, L_y] = i L_z$ and its cyclic permutations. The eigenvalues of $H(\boldsymbol{k})$ are found to be

$$\omega_0(\boldsymbol{k}) = 0, \quad \omega_\pm(\boldsymbol{k}) = \pm \sqrt{c^2 \boldsymbol{k}^2 + (\Omega - \nu_o \boldsymbol{k}^2)^2}, \tag{5}$$

which provide the dispersion relations with the wavevector $\boldsymbol{k}$ of the energy bands. These three dispersion relations are separated by energy gaps resulting from the nonzero chiral body force. We shall refer to $\omega_0(\boldsymbol{k})$ as the flat band and $\omega_\pm(\boldsymbol{k})$ as the Poincaré wave bands, following the shallow water case.

## 2.2 Chern number in continuum fluids

Denoting normalized eigenvectors as $\boldsymbol{\varphi}_0(\boldsymbol{k})$ and $\boldsymbol{\varphi}_\pm(\boldsymbol{k})$ for the energy bands, we can introduce their Berry connections as $\mathcal{A}_\alpha(\boldsymbol{k}) := i\boldsymbol{\varphi}_\alpha^*(\boldsymbol{k}) \cdot \mathrm{d}\boldsymbol{\varphi}_\alpha(\boldsymbol{k})$ for $\alpha = 0, \pm$ (no implicit summation over $\alpha$). We also introduce $\mathcal{C}_\alpha := \int \mathcal{F}_\alpha(\boldsymbol{k})/2\pi$ with the Berry curvature $\mathcal{F}_\alpha(\boldsymbol{k}) := \mathrm{d}\mathcal{A}_\alpha(\boldsymbol{k})$. The integration in the definition of $\mathcal{C}_\alpha$ is performed over the entire wavenumber space. Importantly, since the system is supposed to be in an infinitely extended bulk, its wavenumber space is given as an infinitely extended space, i.e., $\boldsymbol{k} \in \mathbb{R}^2$, which is not compactified. Hence, there is no guarantee that $\mathcal{C}_\alpha$ is quantized, and $\mathcal{C}_\alpha$ cannot be considered as the first Chern number.

Nevertheless, the presence of the nonzero odd viscosity $\nu_o$ allows the wavenumber space to be compactified to $S^2$, thus quantizing the Chern number $\mathcal{C}_\alpha$ [23]. To see this quantization, we consider $N(\boldsymbol{k}) = \boldsymbol{n}(\boldsymbol{k}) \cdot \boldsymbol{L}$ with $\boldsymbol{n}(\boldsymbol{k}) := \boldsymbol{h}(\boldsymbol{k})/|\boldsymbol{h}(\boldsymbol{k})|$, which is a normalized version of $H(\boldsymbol{k})$. Since the eigenvectors of $N(\boldsymbol{k})$ are the same as those of $H(\boldsymbol{k})$, the normalization from $H(\boldsymbol{k})$ to $N(\boldsymbol{k})$ retains the topological properties of $H(\boldsymbol{k})$ associated with the eigenvectors intact. Due to the normalization, $\boldsymbol{n}(\boldsymbol{k})$ takes its value in $S^2$, so that $\boldsymbol{n}$ is considered as a map $\boldsymbol{n} : \mathbb{R}^2 \to S^2$ corresponding to $\boldsymbol{k} \mapsto \boldsymbol{n}(\boldsymbol{k})$. Moreover, when $\nu_o \neq 0$, $\boldsymbol{n}(\boldsymbol{k})$ points toward either the north or south poles of $S^2$ in the limit of $|\boldsymbol{k}| \to \infty$ depending on the sign of $\nu_o$, regardless of the direction of $\boldsymbol{k}$; $\boldsymbol{n}(|\boldsymbol{k}| \to \infty) = [0, 0, -\mathrm{sign}(\nu_o)]^T$. This asymptotic large-$|\boldsymbol{k}|$ behavior of $\boldsymbol{n}(\boldsymbol{k})$ enables us to consistently define the value of $\boldsymbol{n}(\boldsymbol{k})$ at $|\boldsymbol{k}| = \infty$ and to compactify its domain to $S^2$ by adding an infinity point to $\mathbb{R}^2$. Therefore, $\boldsymbol{n}$ can be redefined as a map from $S^2$ to $S^2$, and accordingly, $\mathcal{C}_\alpha$ is guaranteed to be quantized to $2\mathbb{Z}$ as the first Chern number associated with the mapping $\boldsymbol{n} : S^2 \to S^2$. Note that, when $\nu_o = 0$, the large-$|\boldsymbol{k}|$ behavior of $\boldsymbol{n}(\boldsymbol{k})$ depends on the direction of $\boldsymbol{k}$ as $\boldsymbol{n}(|\boldsymbol{k}| \to \infty) = [k_x/|\boldsymbol{k}|, k_y/|\boldsymbol{k}|, 0]^T$, so that the domain of $\boldsymbol{n}$ can not be compactified.

The Chern number for each energy band is indeed calculated as

$$\mathcal{C}_0 = 0, \qquad \mathcal{C}_\pm = \mp\Big[\mathrm{sign}(\Omega) + \mathrm{sign}(\nu_o)\Big]. \tag{6}$$

The Chern numbers are kept constant as far as the non-zero energy gap $\Omega$ is maintained and the compactification of the wavenumber space is preserved with the sign of $\nu_o$. Particularly, even when the non-zero energy gap $\Omega$ is maintained, the Chern numbers $\mathcal{C}_\pm$ of the Poincaré wave bands change with the sign of the odd viscosity.

## 3 Mapping to the gauge theory from hydrodynamics

In order to find a gauge theory corresponding to the Poincaré wave bands, we constrain solutions of Eqs. (1) and (2) in the coordinate space so as to be on the Poincaré wave bands. This constraint can be implemented through local conserved quantities having different values for each energy band, so that we consider the vorticity conservation law

$$\partial_t\Big(\epsilon_{ij}\partial_i v_j(t, \boldsymbol{x})\Big) + \partial_i\Big((\Omega + \nu_o\nabla^2)v_i(t, \boldsymbol{x})\Big) = 0, \tag{7}$$

which is derived from Eq. (2). Combining this vorticity conservation with Eq. (1), we find that $\mathcal{Q}(t, \boldsymbol{x}) := \rho_0 \epsilon_{ij}\partial_i v_j(t, \boldsymbol{x}) - (\Omega + \nu_o\partial_k\partial_k)\rho(t, \boldsymbol{x})$ does not depend on time: $\partial_t\mathcal{Q}(t, \boldsymbol{x}) = 0$. In each energy band, $\mathcal{Q}$ has a constant value reflecting the properties of its eigenvector. In fact, with the use of the normalized eigenvectors $\boldsymbol{\varphi}_\alpha(\boldsymbol{k})$, $\mathcal{Q} = 0$ holds for the Poincaré wave bands and $\mathcal{Q} \propto \omega_\pm(\boldsymbol{k})$ holds for the flat band in the wavenumber space. Therefore, turning it back on the coordinate space, one finds that $\mathcal{Q}(t, \boldsymbol{x}) = 0$ serves as a constraint for the solutions to be on the Poincaré wave band.

In two dimensions, we can introduce U(1) gauge fields $A_\mu(t, \boldsymbol{x})$ for $\mu \in \{t, x, y\}$ so as to match their field strength with hydrodynamic variables. This is because the mass conservation law (1) is automatically satisfied as the Bianchi identity for the U(1) gauge fields under the matching conditions

$$\rho(t, \boldsymbol{x}) = B(t, \boldsymbol{x}), \quad \rho_0 v_i(t, \boldsymbol{x}) = \epsilon_{ij} E_j(t, \boldsymbol{x}), \tag{8}$$

with the field strengths $B(t, \boldsymbol{x}) = \epsilon_{ij} \partial_i A_j(t, \boldsymbol{x})$ and $E_i(t, \boldsymbol{x}) = \partial_i A_t(t, \boldsymbol{x}) - \partial_t A_i(t, \boldsymbol{x})$. In other words, the gauge fields $A_\mu(t, \boldsymbol{x})$ describe a configuration satisfying Eq. (1) without loss of generality. The matching conditions (8) turn Eq. (2) and the constraint $\mathcal{Q}(t, \boldsymbol{x}) = 0$ into

$$\partial_t E_i(t, \boldsymbol{x}) - \epsilon_{ij} c^2 \partial_j B(t, \boldsymbol{x}) = (\Omega + \nu_o \nabla^2) \epsilon_{ij} E_j(t, \boldsymbol{x}), \tag{9}$$
$$\partial_i E_i(t, \boldsymbol{x}) = -(\Omega + \nu_o \nabla^2) B(t, \boldsymbol{x}). \tag{10}$$

Significantly, these equations can be obtained from the variation of the following action with respect to the gauge fields:

$$S_{\text{Poincaré}}[A_\mu] = \frac{1}{2\rho_0} \int dt d^2x \left[ \boldsymbol{E}(t, \boldsymbol{x})^2 - c^2 B(t, \boldsymbol{x})^2 \right.$$
$$\left. - \Omega \epsilon^{\mu\nu\lambda} A_\mu(t, \boldsymbol{x}) \partial_\nu A_\lambda(t, \boldsymbol{x}) - 2\nu_o B(t, \boldsymbol{x}) \partial_k E_k(t, \boldsymbol{x}) \right], \tag{11}$$

which is a Maxwell-Chern-Simons theory with an odd viscosity term. Therefore, the topological waves on the Poincaré wave band of Eqs. (1) and (2) are concisely described by the gauge theory (11). We note that $S_{\text{Poincaré}}[A_\mu]$ can be obtained from writing down a U(1) gauge theory dual to Eqs. (1) and (2) and projecting it onto the Poincaré wave band (See Appendix A).

## 4 BBC in continuum fluids

### 4.1 Edge-mode solutions

We set the boundary at $x = 0$ and assume that the fluid is in $x > 0$ and the region $x < 0$ is empty. We take boundary conditions at $x = 0$ as

$$A_t(t, 0, y) = 0, \quad A_y(t, 0, y) = 0, \quad \partial_x A_t(t, \boldsymbol{x})|_{x=0} = \text{const.}, \quad \partial_x A_y(t, \boldsymbol{x})|_{x=0} = \text{const.} \tag{12}$$

resulting in $E_y(t, 0, y) = 0$ and $\partial_x E_y(t, \boldsymbol{x})|_{x=0} = 0$, i.e., $v_x(t, 0, y) = 0$ and $\partial_x v_x(t, \boldsymbol{x})|_{x=0} = 0$, which means neither the fluid velocity nor its derivative into the boundary exists.[1] From the viewpoint of the gauge theory, this boundary condition $E_y(t, 0, y) = 0$ ensures that the action, in particular the Chern-Simons term, remains gauge invariant in the presence of the boundary. Namely, since the Chern-Simons term is generally gauge invariant up to total derivatives, surface terms at the boundary must vanish to maintain the gauge invariance, which is achieved by $E_y(t, 0, y) = 0$.

---

[1] The boundary conditions for the derivative of the field strength have arbitrariness even under the condition that the system with a boundary preserves the Hermitian property. Depending on a specific choice of the boundary conditions, anomalous edge modes may appear (See Ref. [42]), but we employ the boundary condition (12) for simplicity.

We solve Eqs. (9) and (10) with the boundary conditions (12), whose constants are set to be zero. To drop propagating modes in the bulk, which are irrelevant to our purpose, we take the boundary condition at infinity as

$$A_\mu(t, \boldsymbol{x}) \to 0 \qquad \text{for} \quad x \to \infty. \tag{13}$$

Combining this condition with the boundary condition at $x = 0$, we can set $A_t(t, \boldsymbol{x}) = A_y(t, \boldsymbol{x}) = 0$. Then, the equations of motion read

$$\partial_t^2 A_x(t, \boldsymbol{x}) - c^2 \partial_y^2 A_x(t, \boldsymbol{x}) = 0, \tag{14}$$

$$\partial_t \partial_x A_x(t, \boldsymbol{x}) = -(\Omega + \nu_o \nabla^2) \partial_y A_x(t, \boldsymbol{x}). \tag{15}$$

Taking the ansatz $A_x(t, \boldsymbol{x}) = f(x) e^{i(\omega t - k_y y)}$, we find the dispersion relation of the edge modes as $\omega = \pm c k_y$ from the first equation. These signs correspond to the propagating direction of the modes along the boundary; the mode with $\omega_+ := c k_y$ ($\omega_- := -c k_y$) runs in the direction of increasing (decreasing) $y$. Then, the second equation together with the dispersion relation turns into

$$(-\tilde{\nu}_o \partial_X^2 \pm \partial_X) \tilde{f}_\pm(X) = (1 - \tilde{\nu}_o \tilde{k}^2) \tilde{f}_\pm(X), \tag{16}$$

with double sign $\pm$ being in the same order.[2] Here, we have introduced dimensionless coordinate $X := \Omega x / c$, wavenumber $\tilde{k} := c k_y / \Omega$, and parameter $\tilde{\nu}_o := \Omega \nu_o / c^2$. The function $\tilde{f}_\pm(X) := f(x)$ corresponds to the edge mode with $\omega_\pm$ for each sign of the index.

The general solution for $\tilde{f}_-(X)$ is given by a linear summation of $e^{\kappa_-^{(1)} X}$ and $e^{\kappa_-^{(2)} X}$ with

$$\kappa_-^{(1)} = \frac{-1 + \sqrt{1 - 4\tilde{\nu}_o + 4\tilde{\nu}_o^2 \tilde{k}^2}}{2\tilde{\nu}_o}, \qquad \kappa_-^{(2)} = \frac{-1 - \sqrt{1 - 4\tilde{\nu}_o + 4\tilde{\nu}_o^2 \tilde{k}^2}}{2\tilde{\nu}_o}. \tag{17}$$

Similarly, the general solution for $\tilde{f}_+(X)$ is given by a linear summation of $e^{\kappa_+^{(1)} X}$ and $e^{\kappa_+^{(2)} X}$ with

$$\kappa_+^{(1)} = -\kappa_-^{(1)}, \qquad \kappa_+^{(2)} = -\kappa_-^{(2)}. \tag{18}$$

The solutions $f_\pm(X)$ have to satisfy the boundary condition (13), so that the condition $\text{Re}[\kappa] < 0$ is imposed on the exponents $\kappa$. At $\tilde{\nu}_o = 0$, realized in the limit of $\nu_o \to 0$ with keeping $\Omega$ nonzero, only $\kappa_-^{(1)}$ is negative and remains finite. Indeed, the solution with $\kappa_-^{(1)}$ corresponds to the so-called coastal Kelvin wave of shallow water equations describing the oceans on Earth in the context of geophysical fluids [47].

## 4.2    Number of the edge modes

As we can see in Fig. 1, where the real parts of the exponents are plotted as functions of $\tilde{k}$, the exponents satisfying the condition $\text{Re}[\kappa] < 0$ differ across the threshold wavenumber $\tilde{k}_c \equiv 1/\sqrt{\tilde{\nu}_o}$. Here, $\tilde{k}_c$ makes the square root of the exponent zero; $1 - 4\tilde{\nu}_o + 4\tilde{\nu}_o^2 \tilde{k}_c^2 = 0$. Accordingly, the normalizable solutions switch at the threshold wavenumber $\pm \tilde{k}_c$ and they are summarized as

$$f_-(X) = c_1 e^{\kappa_-^{(1)} X} + c_2 e^{\kappa_-^{(2)} X}, \qquad f_+(X) = 0, \qquad \text{for} \quad |\tilde{k}| < \tilde{k}_c, \tag{19a}$$

$$f_-(X) = c_3 e^{\kappa_-^{(2)} X}, \qquad f_+(X) = c_4 e^{\kappa_+^{(1)} X}, \qquad \text{for} \quad |\tilde{k}| > \tilde{k}_c, \tag{19b}$$

for $\nu_o > 0$ and

$$f_-(X) = c_5 e^{\kappa_-^{(1)} X}, \quad f_+(X) = c_6 e^{\kappa_+^{(2)} X}, \qquad \text{for any } \tilde{k}, \tag{20}$$

for $\nu_o < 0$ with $c_i$ being arbitrary constants.

---

[2]The equation (9) in the $y$ direction together with $\omega_\pm = \pm c k_y$ provides Eq. (16).

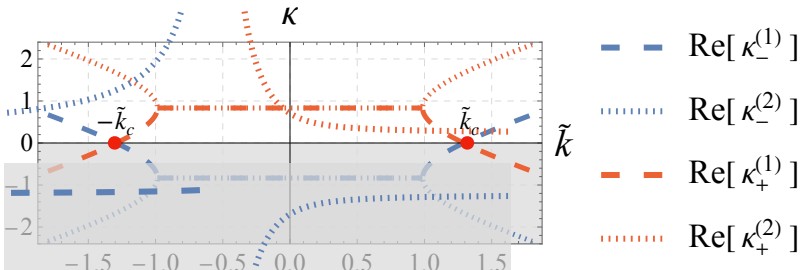

Figure 1: Plot of the real parts of the exponents $\kappa_{\pm}^{(1)}$ and $\kappa_{\pm}^{(2)}$ as functions of $\tilde{k}$ for $\tilde{\nu}_o = 0.8$. The shading represents the region where the exponents satisfy the normalizability condition $\mathrm{Re}[\kappa] < 0$ for solutions in $x > 0$. At $\tilde{k} = \pm\tilde{k}_c = \pm 1/\sqrt{\tilde{\nu}_o}$, indicated by the red dots, the exponents $\kappa_{\pm}^{(1)}$ becomes zero.

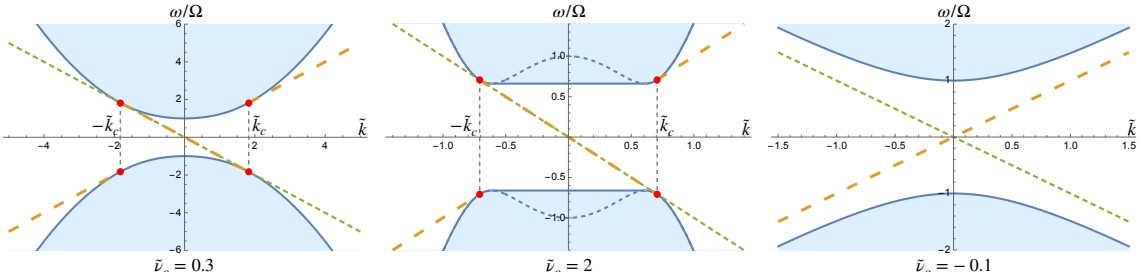

Figure 2: Plot of the energy bands including the dispersion relation of the edge modes from Eqs. (19) and (20) for $\tilde{\nu}_o = 0.3$ (Left), $\tilde{\nu}_o = 2$ (Middle), and $\tilde{\nu}_o = -0.1$ (Right). While the bulk energy band is depicted by the blue shaded regime, the dispersions of the edge modes are represented by the green and orange dashed lines. (Left and Middle) When $\tilde{\nu}_o > 0$, there are two modes with $\omega/\Omega = -\tilde{k}$ in $|\tilde{k}| < \tilde{k}_c$, and one mode with $\omega/\Omega = -\tilde{k}$ and one mode $\omega/\Omega = \tilde{k}$ in $|\tilde{k}| > \tilde{k}_c$. The edge mode dispersions touch the bulk energy bands at $|\tilde{k}| = \tilde{k}_c$. (Right) When $\tilde{\nu}_o < 0$, there are one mode with $\omega/\Omega = -\tilde{k}$ and one mode $\omega/\Omega = \tilde{k}$ for any $\tilde{k}$, and these edge mode dispersions do not touch the bulk energy bands.

From Eqs. (19) and (20), the energy bands including the dispersion relation of the edge modes can be shown as in Fig. 2. There, the dispersion branches of the edge modes, represented by the green and orange dashed lines, exist up to infinitely large wavenumbers without being absorbed into the bulk energy band. Such edge modes generally appear in continuous fluids because their wavenumber space has no periodicity. In particular, the dispersion branches of edge modes, which can extend to infinitely large wavenumbers, sometimes enter the gapped region. For this reason, we cannot count them as the number of intersections with a horizontal line placed within the gap, as is commonly done in solid-state physics. Let us define the effective net number $\mathcal{N}_{\mathrm{edge}}$ of the chiral edge modes for the lower band in an unconventional way [42] as

$$\mathcal{N}_{\mathrm{edge}} \equiv \mathcal{N}_{\mathrm{edge};-} - \mathcal{N}_{\mathrm{edge};+}, \tag{21}$$

where $\mathcal{N}_{\mathrm{edge};-}$ ($\mathcal{N}_{\mathrm{edge};+}$) is the number of dispersion branches of edge modes that are absorbed into (released from) the lower bulk energy band as the wavenumber increases. This definition allows us to count the number of edge modes properly not only in crystals but also in continuum fluids. Indeed, $\mathcal{N}_{\mathrm{edge};-}$ and $\mathcal{N}_{\mathrm{edge};+}$ are 3 and 1, respectively, for $\tilde{\nu}_o > 0$, while both are zero for $\tilde{\nu}_o < 0$. Therefore, $\mathcal{N}_{\mathrm{edge}}$ is equal to $\mathcal{C}_-$ for both $\tilde{\nu}_o > 0$ and $\tilde{\nu}_o < 0$, and the BBC still holds at least in this sense.

# 5   Discussions and outlooks

In summary, we have mapped the dynamics of the topological waves in continuum fluids with odd viscosity, described by Eqs. (1) and (2), into a gauge theory via the matching condition (8) and the constraint $\mathcal{Q}(t, \boldsymbol{x}) = 0$ onto the Poincare wave band. Specifically, the equations of motion for the mapped gauge fields [Eqs. (9) and (10)] are obtained from the Maxwell-Chern-Simons theory with an additional term associated with the odd viscosity [Eq. (11)]. We have provided a direct mapping of the equations of motion in Sec. 3, while we have also discussed the derivation of the gauge action from the hydrodynamic equations by using the projection onto the Poincaré wave band in Appendix A.

We have then investigated chiral edge modes within the obtained gauge theory. Setting the boundary at $x = 0$ and taking the boundary conditions (12) and (13), we have solved the equations of motions and found the edge mode solutions [Eqs. (19) and (20)]. From the corresponding band structure in Fig. 2, we have argued that the BBC still holds even in continuous fluids with nonzero odd viscosity if the effective number of edge modes is defined in an appropriate manner.

In Fig. 2, the threshold wavenumber $\tilde{k}_c = 1/\sqrt{\tilde{\nu}_o}$ plays an important role for topological waves associated with the odd viscosity because $\tilde{k}_c$ and $-\tilde{k}_c$ give the endpoints of the dispersion branches of the edge modes for $\tilde{\nu}_o > 0$. As one can see in Eqs. (9) and (10), the chiral body force term and the odd viscosity term can be expressed in a unified manner by the combination, $(\Omega + \nu_o \nabla^2)$. Thus, if we regard $(\Omega + \nu_o \nabla^2)$ as the strength of an effective chiral body force, its sign is reversed at $\tilde{k} = \pm \tilde{k}_c$, or equivalently $\Omega = \nu_o k_y^2$. This sign reversal results in an inversion of the propagation direction of the edge mode, while maintaining the non-zero energy gap. In fact, in Fig. 1, the signs of $\mathrm{Re}[\kappa_-^{(1)}]$ and $\mathrm{Re}[\kappa_+^{(1)}]$ are inverted at $\tilde{k} = \pm \tilde{k}_c$, so that the edge mode propagating with $\omega = -c k_y$ in $|\tilde{k}| < \tilde{k}_c$ changes to the opposite propagation with $\omega = +c k_y$ in $|\tilde{k}| > \tilde{k}_c$.

With the help of the matching conditions (8), we have provided a U(1) gauge theory (11) for topological waves on the Poincaré wave band. Beyond the analogy between the matrix $H(\boldsymbol{k}) = \boldsymbol{h}(\boldsymbol{k}) \cdot \boldsymbol{L}$ and the two-dimensional massive Dirac Hamiltonian, this gauge theory establishes a correspondence between quantum Hall states and topological waves induced by the odd viscosity. In quantum Hall states, U(1) gauge theories with the Chern-Simons term were written down as a low-energy effective theory for gapped Landau levels, and the odd viscosity term $B \nabla \cdot \boldsymbol{E}$ was found [48–54]. This correspondence without odd viscosity has been discussed in Ref. [55], and our results clarify the role of the odd viscosity, in particular, for the edge modes. Topological waves in classical fluids have been discussed as having a field-theoretic correspondence with quantum Hall states, in a manner distinct from that in [55], which could help one to understand their topological nature [56–58]. While we have identified edge modes from U(1) gauge theory, we have not discussed the Chern number from a field-theoretic point of view; we leave them for future work.

## Acknowledgments

The authors thank Kazuki Sone and Tomoki Ozawa for useful discussions.

**Funding information**   K.F. is supported by JSPS KAKENHI Grant Number JP24KJ0062. Y.A. acknowledges support from JSPS Grant Number JP19K23424 and from JST FOR-EST Program (Grant No. JPMJFR222U, Japan).

# A    Derivation of the gauge theory

In this appendix, we derive the gauge action (11) from the hydrodynamic equations (1) and (2).

## A.1    Hydrodynamic action and its dual gauge action

Applying the method to incorporate the chiral body force into a field theory [55], we start with the following action providing the equations of motion (1) and (2):

$$
S_{\text{hydro}}[\rho, v_i, \theta, p, q, p_j, q_j] = \int dt d^2 x \left[ \frac{\rho_0}{2} \boldsymbol{v}(t, \boldsymbol{x})^2 - \frac{c^2}{2\rho_0} \rho(t, \boldsymbol{x})^2 + \theta(t, \boldsymbol{x}) \Big( \partial_t \rho(t, \boldsymbol{x}) + \rho_0 \partial_i v_i(t, \boldsymbol{x}) \Big) \right.
$$

$$
+ \rho_0 p(t, \boldsymbol{x}) \partial_t q(t, \boldsymbol{x}) - \rho_0 p_k(t, \boldsymbol{x}) \partial_t q_k(t, \boldsymbol{x}) - \rho_0 v_i(t, \boldsymbol{x}) \Big( p(t, \boldsymbol{x}) \partial_i \beta(\boldsymbol{x}) - q(t, \boldsymbol{x}) \partial_i \alpha(\boldsymbol{x}) \Big)
$$

$$
\left. + \rho_0 \partial_k v_i(t, \boldsymbol{x}) \Big( p_k(t, \boldsymbol{x}) \partial_i \beta_o(\boldsymbol{x}) - q_k(t, \boldsymbol{x}) \partial_i \alpha_o(\boldsymbol{x}) \Big) \right], \tag{22}
$$

where $\theta(t, \boldsymbol{x})$ is an auxiliary field to impose mass continuity and $p(t, \boldsymbol{x}), q(t, \boldsymbol{x}), p_j(t, \boldsymbol{x})$, and $q_j(t, \boldsymbol{x})$ are auxiliary fields to describe the chiral body force and the odd viscosity. Here, $\beta(\boldsymbol{x}), \alpha(\boldsymbol{x}), \beta_o(\boldsymbol{x})$, and $\alpha_o(\boldsymbol{x})$ are background fields related with $\Omega$ and $\nu_o$ as

$$
\Omega = \epsilon^{ij} \partial_i \beta(\boldsymbol{x}) \partial_j \alpha(\boldsymbol{x}), \qquad \nu_o = \epsilon^{ij} \partial_i \beta_o(\boldsymbol{x}) \partial_j \alpha_o(\boldsymbol{x}). \tag{23}
$$

Taking variations of the action $S_{\text{hydro}}$ with respect to the fields leads to the equations of motion as

$$
\rho(t, \boldsymbol{x}) = -\frac{\rho_0}{c^2} \partial_t \theta(t, \boldsymbol{x}), \tag{24}
$$

$$
v_i(t, \boldsymbol{x}) = \partial_i \theta(t, \boldsymbol{x}) + \Big( p(t, \boldsymbol{x}) \partial_i \beta(\boldsymbol{x}) - q(t, \boldsymbol{x}) \partial_i \alpha(\boldsymbol{x}) \Big),
$$

$$
+ \partial_k \Big( p_k(t, \boldsymbol{x}) \partial_i \beta_o(\boldsymbol{x}) - q_k(t, \boldsymbol{x}) \partial_i \alpha_o(\boldsymbol{x}) \Big) \tag{25}
$$

$$
\partial_t \rho(t, \boldsymbol{x}) + \rho_0 \partial_i v_i(t, \boldsymbol{x}) = 0, \tag{26}
$$

and

$$
\partial_t q(t, \boldsymbol{x}) = v_i(t, \boldsymbol{x}) \partial_i \beta(\boldsymbol{x}), \qquad \partial_t p(t, \boldsymbol{x}) = v_i(t, \boldsymbol{x}) \partial_i \alpha(\boldsymbol{x}), \tag{27}
$$

$$
\partial_t q_k(t, \boldsymbol{x}) = \partial_k v_i(t, \boldsymbol{x}) \partial_i \beta_o(\boldsymbol{x}), \qquad \partial_t p_k(t, \boldsymbol{x}) = \partial_k v_i(t, \boldsymbol{x}) \partial_i \alpha_o(\boldsymbol{x}). \tag{28}
$$

Combining these equations of motion, one can get

$$
\partial_t v_i(t, \boldsymbol{x}) + \frac{c^2}{\rho_0} \partial_i v_i(t, \boldsymbol{x}) - \nu_o \epsilon_{ij} \nabla^2 v_j(t, \boldsymbol{x}) = \Omega \epsilon_{ij} v_j(t, \boldsymbol{x}). \tag{29}
$$

Therefore, as closed equations for the hydrodynamic variables $\rho(t, \boldsymbol{x})$ and $v_i(t, \boldsymbol{x})$, we find the hydrodynamic equations (26) and (29).

With the use of the matching conditions (8), we can write down a U(1) gauge theory dual to $S_{\text{hydro}}$ as

$$
S_{\text{dual}}[A_\mu, p, q, p_j, q_j] = \int dt d^2 x \left[ \frac{1}{2\rho_0} \boldsymbol{E}(t, \boldsymbol{x})^2 - \frac{c^2}{2\rho_0} B(t, \boldsymbol{x})^2 \right.
$$

$$
+ \rho_0 p(t, \boldsymbol{x}) \partial_t q(t, \boldsymbol{x}) - \rho_0 p_k(t, \boldsymbol{x}) \partial_t q_k(t, \boldsymbol{x}) + \epsilon_{ij} E_i(t, \boldsymbol{x}) \Big( p(t, \boldsymbol{x}) \partial_j \beta(\boldsymbol{x}) - q(t, \boldsymbol{x}) \partial_j \alpha(\boldsymbol{x}) \Big)
$$

$$
\left. - \epsilon_{ij} \partial_k E_i(t, \boldsymbol{x}) \Big( p_k(t, \boldsymbol{x}) \partial_j \beta_o(\boldsymbol{x}) - q_k(t, \boldsymbol{x}) \partial_j \alpha_o(\boldsymbol{x}) \Big) \right]. \tag{30}
$$

Here, since the mass conservation always holds as the Bianchi identity, we can drop the auxiliary field $\theta(t, \boldsymbol{x})$ and the terms related to it.

For later use, we note the equations of motion from the variation of $S_{\text{dual}}$ with respect to the fields as

$$\partial_i E_i(t, \boldsymbol{x}) = -\rho_0 \epsilon_{ij} \Big( \partial_i p(t, \boldsymbol{x}) \partial_j \beta(\boldsymbol{x}) - \partial_i q(t, \boldsymbol{x}) \partial_j \alpha(\boldsymbol{x}) \Big)$$
$$- \rho_0 \epsilon_{ij} \partial_k \Big( \partial_i p_k(t, \boldsymbol{x}) \partial_j \beta_o(\boldsymbol{x}) - \partial_i q_k(t, \boldsymbol{x}) \partial_j \alpha_o(\boldsymbol{x}) \Big), \quad (31)$$

$$\partial_t E_i(t, \boldsymbol{x}) - \epsilon^{ij} c^2 \partial_j B(t, \boldsymbol{x}) = -\rho_0 \epsilon_{ij} \Big( \partial_t p(t, \boldsymbol{x}) \partial_j \beta(\boldsymbol{x}) - \partial_t q(t, \boldsymbol{x}) \partial_j \alpha(\boldsymbol{x}) \Big)$$
$$- \rho_0 \epsilon_{ij} \partial_k \Big( \partial_t p_k(t, \boldsymbol{x}) \partial_j \beta_o(\boldsymbol{x}) - \partial_t q_k(t, \boldsymbol{x}) \partial_j \alpha_o(\boldsymbol{x}) \Big), \quad (32)$$

and

$$\rho_0 \partial_t q(t, \boldsymbol{x}) = -\epsilon_{ij} E_i(t, \boldsymbol{x}) \partial_j \beta(\boldsymbol{x}), \qquad \rho_0 \partial_t p(t, \boldsymbol{x}) = -\epsilon_{ij} E_i(t, \boldsymbol{x}) \partial_j \alpha(\boldsymbol{x}), \quad (33)$$
$$\rho_0 \partial_t q_k(t, \boldsymbol{x}) = -\epsilon_{ij} \partial_k E_i(t, \boldsymbol{x}) \partial_j \beta_o(\boldsymbol{x}), \qquad \rho_0 \partial_t p_k(t, \boldsymbol{x}) = -\epsilon_{ij} \partial_k E_i(t, \boldsymbol{x}) \partial_j \alpha_o(\boldsymbol{x}). \quad (34)$$

## A.2  Projection onto the Poincaré wave band

The constraint $\mathcal{Q}(t, \boldsymbol{x}) = 0$ on the solutions of the Poincaré wave band is expressed in terms of the gauge fields as

$$\partial_i E_i(t, \boldsymbol{x}) + (\Omega + \nu_o \partial_k \partial_k) B(t, \boldsymbol{x}) = 0. \quad (35)$$

Combining this constraint with the equation of motion (31), one finds

$$(\Omega + \nu_o \partial_k \partial_k) B(t, \boldsymbol{x}) = \rho_0 \epsilon_{ij} \Big( \partial_i p(t, \boldsymbol{x}) \partial_j \beta(\boldsymbol{x}) - \partial_i q(t, \boldsymbol{x}) \partial_j \alpha(\boldsymbol{x}) \Big)$$
$$+ \rho_0 \epsilon_{ij} \partial_k \Big( \partial_i p_k(t, \boldsymbol{x}) \partial_j \beta_o(\boldsymbol{x}) - \partial_i q_k(t, \boldsymbol{x}) \partial_j \alpha_o(\boldsymbol{x}) \Big), \quad (36)$$

and accordingly

$$(\Omega + \nu_o \partial_k \partial_k) A_j(t, \boldsymbol{x}) = \rho_0 \Big( p(t, \boldsymbol{x}) \partial_j \beta(\boldsymbol{x}) - q(t, \boldsymbol{x}) \partial_j \alpha(\boldsymbol{x}) \Big)$$
$$+ \rho_0 \partial_k \Big( p_k(t, \boldsymbol{x}) \partial_j \beta_o(\boldsymbol{x}) - q_k(t, \boldsymbol{x}) \partial_j \alpha_o(\boldsymbol{x}) \Big), \quad (37)$$

where we dropped irrelevant pure gauge factor. This equation allows us to express the auxiliary fields in terms of $A_i(t, \boldsymbol{x})$:

$$p(t, \boldsymbol{x}) = \frac{1}{\rho_0} \epsilon_{ij} A_i(t, \boldsymbol{x}) \partial_j \alpha(\boldsymbol{x}), \qquad q(t, \boldsymbol{x}) = \frac{1}{\rho_0} \epsilon_{ij} A_i(t, \boldsymbol{x}) \partial_j \beta(\boldsymbol{x}), \quad (38)$$

and

$$p_k(t, \boldsymbol{x}) = \frac{1}{\rho_0} \epsilon_{ij} \partial_k A_i(t, \boldsymbol{x}) \partial_j \alpha_o(\boldsymbol{x}), \qquad q_k(t, \boldsymbol{x}) = \frac{1}{\rho_0} \epsilon_{ij} \partial_k A_i(t, \boldsymbol{x}) \partial_j \beta_o(\boldsymbol{x}). \quad (39)$$

Substituting these expressions into the action $S_{\text{dual}}$, we obtain

$$S_{\text{Poincaré}}[A_\mu] = \frac{1}{2\rho_0} \int dt d^2 x \Big[ \boldsymbol{E}(t, \boldsymbol{x})^2 - c^2 B(t, \boldsymbol{x})^2 - \Omega \epsilon^{\mu\nu\lambda} A_\mu(t, \boldsymbol{x}) \partial_\nu A_\lambda(t, \boldsymbol{x})$$
$$- 2\nu_o B(t, \boldsymbol{x}) \partial_k E_k(t, \boldsymbol{x}) \Big]. \quad (40)$$

As a result, imposing the condition (35) has allowed us to integrate out the auxiliary fields from the action $S_{\text{dual}}$, yielding the action $S_{\text{Poincaré}}$ for the gauge fields $A_\mu$ alone. While the first three terms of the right-hand side, i.e., the Maxwell terms and the Chern-Simons term, can be found in Ref. [55], the last term is new.

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
