# Peer review of "Gauge theory for topological waves in continuum fluids with odd viscosity"

_SciPost Physics_

## Round 1 · Referee Report · Anonymous (Referee 1) · 2024-12-10

Strengths

  1. Clearly written
  2. Gives necessary literature references

Weaknesses

  1. Does not sufficiently advance knowledge

Report

I cannot recommend this manuscript for the publication in SciPost Physics.

The manuscript presents a mapping of the fluid dynamics equations with odd viscosity to 2+1 Maxwell-Chern-Simons equations with an additional term corresponding to odd viscosity. This mapping is known (see Ref. [55]). The only new term that is added is the one corresponding the odd viscosity. This is a higher gradient term that is very straightforward to write using the mapping of [55].

The authors use the obtained gauge theory to study the bulk-boundary correspondence. However, this can be done directly in the fluid formulation as it was shown in [23,42] and in [56-58]. The bulk and edge dispersions have also been derived in those papers. I do not think that using the mapping to the gauge fields helps to understand the bulk-boundary correspondence of hydrodynamic modes better.

Requested changes

There are confusions between space and reciprocal space at a few points. The following sentences are simply wrong.

"As a crucial difference from solid crystals, continuum fluids lack the spatial periodicity. " (an actual difference is that the spatial periodicity in fluids is continuous, not discrete as in crystals)

"Importantly, since the system is supposed to be in an infinitely extended bulk, its wavenumber space is given as an infinitely extended space, ..." (wrong, the reason for an infinite wavenumber space is not the infinitely extended bulk, but the fact that the translational period is zero)

Recommendation

Reject

---

## Round 1 · Referee Report · Anonymous (Referee 2) · 2025-1-28

Report

A couple of years ago, a gauge theoretic formulation of the shallow water equations was constructed. The waves in this system are governed by a Chern-Simons theory, which providing a field-theoretical explanation for the chiral edge modes which had previously been shown to have a topological underpinning.

This paper extends the gauge theory for shallow water and by adding a term corresponding to Hall viscosity. This is a well-studied, non-dissipative viscosity, lending itself to an action principle.

The Hall viscosity has also been well-studied in the context of shallow water. It was previously used as a kind of UV regulator when computing Chern numbers in this system. So the paper really translates this term into a gauge theory variables.

The paper is right. But I'm not convinced that the results are that important. The Hall viscosity term is rather obvious and I couldn't see any particular surprises in the results.

For this reason, I'm not convinced that this paper meets the high bar for publication in SciPost. It may well be that SciPost Core is a more appropriate journal.

Recommendation

Accept in alternative Journal (see Report)

---

## Editorial Decision

resubmitted